# The Effects of *Angelica ternata* Extract from Kyrgyzstan on the Formation of *Candida albicans* ATCC 10231 Biofilms

Nadezhda Sachivkina [1,*], Arfenya Karamyan [1], Valentina Semenova [1], Aleksej Ignatiev [1], Abdugani Abdurasulov [2], Rakhima Muratova [2], Dinara Emilbekova [2], Venera Ermatova [2], Ali Risvanli [3], Ruslan Salykov [3], Alfia Ibragimova [1] and Ekaterina Neborak [1]

[1] RUDN University Named after Patrice Lumumba, 117198 Moscow, Russia; arfenya@mail.ru (A.K.); semenova_vi@pfur.ru (V.S.); 1142220883@rudn.ru (A.I.); ibragimova-an@rudn.ru (A.I.); neborak_ev@pfur.ru (E.N.)

[2] Osh State University, Osh 723500, Kyrgyzstan; abdurasul65@mail.ru (A.A.); miss.rakhima@mail.ru (R.M.); dinara-metodica@mail.ru (D.E.); ermatova_v@mail.ru (V.E.)

[3] Kyrgyz-Turkish Manas University, Bishkek 720044, Kyrgyzstan; ali.risvanli@manas.edu.kg (A.R.); salykov.1958@mail.ru (R.S.)

\* Correspondence: sachivkina@yandex.ru

**Abstract:** The therapeutic potential of *Angelica ternata* extract was tested against biofilm-associated fungi *Candida albicans*. Such an extract with a 1.896 ± 0.071% per 1 mL rutin content of flavonoids can reliably suppress the formation of biofilms of pathogenic yeast-like fungi up to 76.6%. The herbal medicine has a significant effect on the initiation, adhesion, and development of biofilms. If the extract is added to a developed biofilm, it has a significant effect on the matrix. As a result, the "glue" between the cells is washed out and they are more easily washed away from the well of the microplate to which they are attached. In this case, the optical density of the biofilm is halved (50.5%).

**Keywords:** *Candida albicans*; biofilms; extracellular matrix; *Angelica ternate*; flavonoids





## 1. Introduction

Candidiasis is frequently involved in surface-associated biofilm-formation disease [1,2]. These networks have multifaceted interactions with the host. Biofilms may be considered as aggregates of microbes enmeshed in an extracellular matrix (ECM) consisting of multifarious polymeric components, forming a complex three-dimensional architecture on biotic and abiotic surfaces [3,4]. ECM acts as a glue, surrounding and clasping together cells in biofilms. Of note, *Candida* infections are commonly associated with biofilms that can form either on mucosal surfaces or on plastic surfaces of indwelling devices as well [5]. The matrix mannan–glucan complex in biofilm is conserved across the *Candida* genus, including *Candida albicans*, *C. tropicalis,* and *C. glabrata* [6,7]. Biofilms formed by a variety of *Candida* spp. tend to vary in density and morphology of the cells themselves (larger or smaller; round or elliptical) and their hyphal forms [8–10]. The biofilm structures contain a heterogeneous polymeric extracellular matrix, providing a protective encasement for the fungal cells [11,12]. Pathogenic yeast-like fungi (YLF), in general, proliferate as adherent biofilms, and the aggregated communities offer resistance to antifungals and host immune responses, rendering them difficult to treat or eradicate. Biofilms have the potential to modulate host immunity throughout various developmental stages [13,14]. During mature biofilm formation, the ECM contributes to resistance to host defenses, and with the dispersal of fungal cells, a more virulent phenotype might appear to aggravate the pathogenesis [1,6,15]. The wide prevalence of *Candida albicans*, the ability to quickly adapt to changing environmental conditions and resistance to antifungal drugs and disinfectants, including due to good communication and cooperation with other microorganisms and the interweaving of biofilms, allow us to consider such YLF as a dangerous infectious agent capable of causing many diseases both in human and veterinary medicine [16–19].

At present, according to scientific publications, great attention is paid to antimycotic preparations from plants that enhance the effect of existing drugs on the pharmaceutical market [20–24]. This is due to several factors. First, they are safe for animals and humans and are non-toxic. Secondly, developing such a "supplement" is much cheaper than researching, bringing to market, and producing a new antibiotic. Thirdly, such "supplements" are easy to replace in case of development of resistance to them. Several studies by our team have shown that some herbal extracts have proven their effectiveness in vitro and in vivo [25–29].

This year, a team from RUDN University was lucky to take part in an international epizootiological expedition, which was carried out by staff of Osh State University in different regions of Kyrgyzstan. During the expedition, we talked a lot with the locals, who know a lot about medicinal herbs. This is due to the fact that the locals live in yurts high in the mountains and medical care is difficult for them because of long distances to the nearest cities. Therefore, much attention is paid to traditional medicine and herbal medicine. The locals told and showed us a plant that they call «Archorot», which is often used to treat people and animals with various infectious diseases. They use this plant in the form of a decoction or feed it to farm animals in its raw form. We found out the scientific name of this plant—*Angelica ternate.*

According to Pubmed, only two articles by Chinese scientists are devoted to this particular type of plant [30,31]. However, thousands of works, especially those of Chinese origin, are devoted to the related plant species—*Angelica sinensis* [32–35], *Angelica gigas* [36–38], and *Angelica dahurica* [39–41], and we have provided links to the articles that are closest to our topic.

Considering all of the above we decided to investigate the effect of *Angelica ternate* extract, determine the number of flavonoids in it, and see how this extract will act on one of the main pathogens according to the WHO—*Candida albicans*—namely on its cells, biofilms, and extracellular matrix.

## 2. Materials and Methods

### 2.1. Plant Collection

The *Angelica ternata* (AT) plant collection (leaves, inflorescences, and stems) was effectuated during international scientific and practical expedition "Traditions and innovations in the development of the agricultural sector of the mountainous regions of Kyrgyzstan", which took place from 10 to 19 July 2023 and was organized by Osh State University (Figure 1). The collection of plants was carried out according to the coordinates 39°40′37″ northern latitude 71°48′15″ east longitude, and at an altitude above sea level 3040 m. After air drying, samples were packed and sent to the Microbiology laboratory of RUDN University.

AT is a biennial herbaceous plant with a radish, massive rhizome. The stem is erect, up to 150 cm high, furrowed, green, and in the lower part painted in purple. The leaves are alternate, with expanded sheaths, twice—thrice pinnate. There is a compound umbel at the end of the stem without wrap. Its flowers are small with visible sepals. Its five petals are elliptical, curved inwards, and greenish-yellow. The fruit is ovoid, flattened from the sides, splits into two winged halves, and naked, with a strong characteristic odor. AT is distributed in the territory of Central Asia in damp places, along the banks of mountain rivers. Angelica herb is rich in proteins, fats, and fiber, containing up to 23% of sugars. AT essential oil contains phellandrene, terpenes, valeric acid, sesquiterpenoids, phytosterols, phthalides, and polyacetylenic compounds. AT plant contains vitamin C, bergapten, emperorin, archicin, and coumarin derivatives [34]. Young leaves, stems, petioles, and flowers are eaten both fresh and as infusions. The powder of angelica flowers is brewed as a tea and used as a spice. The plant is widely used in traditional medicine.

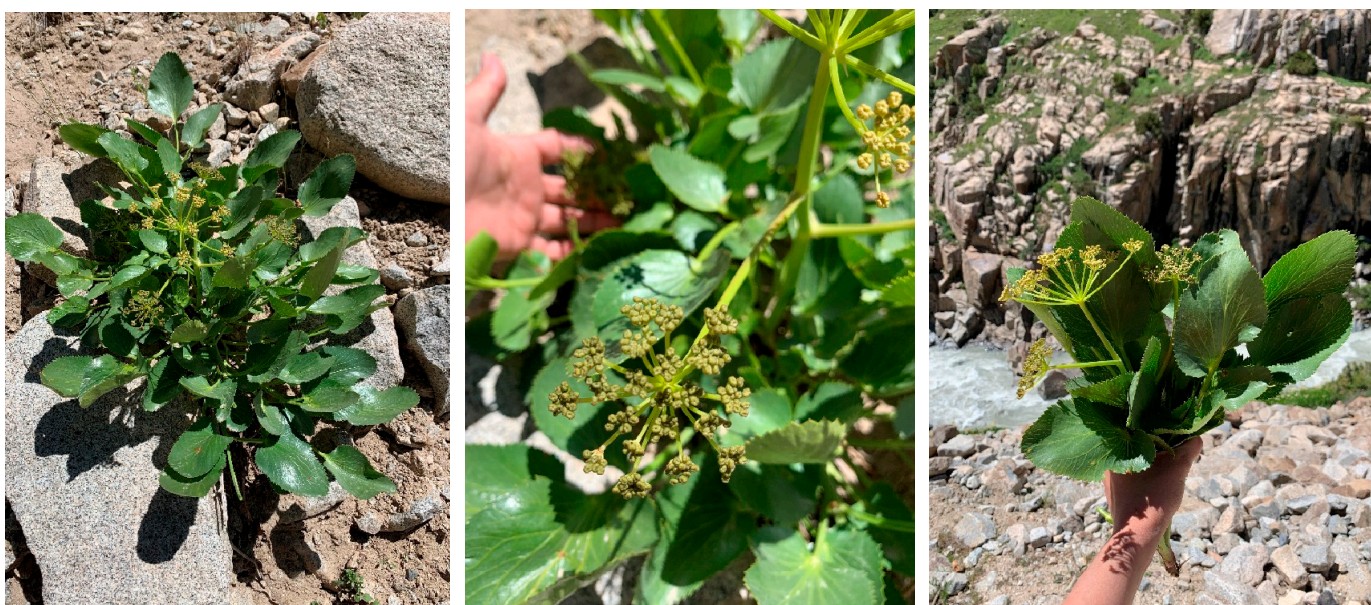

**Figure 1.** Collection of plants in Kyrgyzstan.

In Russian folk medicine, AT is used as diaphoretic, diuretic, and carminative. Root tea is drank for indigestion. A quarter of a teaspoon of the powder of the roots is drank for urinary retention and pneumonia. Alcohol tincture and root decoctions are used externally as rubbing and baths for gout, rheumatism, and lumbar pain. In Chinese folk medicine, Angelica herb is used as a hemostatic, analgesic, and antispasmodic agent [35].

### 2.2. Angelica Ternata Extract Preparation

The plant material was dried outdoors in the shade for 7 days. Subsequently, the dry plants were crushed into 0.5 cm pieces. The extraction of phytochemical compounds was performed with 80% hydroethanolic solution as solvent (270 mL) with thirty grams (30 g) of the vegetal material. The mixture was stirred at 300 rpm at 25 °C for 24 h, thereafter filtration was performed using Whatman filter paper N° 1. The filtered extract was further concentrated at 40 °C in a rotary evaporator (IKA RV8, Staufen, Germany) equipped with a water bath IKA HB10 (IKA Werke, Staufen, Germany) and a vacuum pumping unit IKA MVP10 (IKA Werke, Staufen, Germany). Prepared *Angelica ternata* extract (ATE) was stored in a refrigerator at +4 °C for no more than 7 days. To avoid significant losses, the extract was recovered when the remaining volume was small enough (about 10 mL).

### 2.3. Quantitative Determination of the Amount of Flavonoids in ATE

An analytical sample of the raw material was crushed to the size of particles passing through a sieve with openings of 2 mm in diameter. Then, 3.0 g of crushed raw materials were placed in a flask with a 250 mL section and 100 mL of 80% ethanol were added. The flask was attached to a reflux condenser and heated in a boiling water bath for 60 min, periodically shaking to wash the particles of the raw material from the walls. The hot extract was cooled and filtered through a Filtrak paper filter (Technocom, Yekaterinburg, Russia) into a volumetric flask with a capacity of 100 mL so that the raw material particles did not fall on the filter. After cooling, the filter was washed with 80% ethanol and the extraction volume was adjusted to the mark and mixed (solution A). Solution A (2 mL) was placed into a 25 mL volumetric flask, and added to it was 5 mL of a 2% aluminum chloride solution in 95% ethanol, 1 drop of 5% acetic acid; this brought the volume of the solution to the mark with 95% ethanol (solution B). After 15 min, the optical density of the solution was measured on an SF-2000 spectrophotometer at a wavelength of 410 nm in a cell with a layer thickness of 10 mm. This solution was used as a reference solution. The content of the

total flavonoids in terms of rutin-standard for absolutely dry raw materials in percent (X) was calculated by the formula:

$$X = \frac{A \bullet ao \bullet 100 \bullet P}{Ao \bullet a \bullet (100 - W)} \quad (1)$$

where A—optical density (solution B) of the test solution; $A_o$—optical density (solution B) of rutin standard sample (RSS); a—weighed raw materials in grams; a—sample of RSS in grams; P—the content of the main substance in the RSS in %; and W—the weight loss on drying (%). Preparation of RSS: 0.05 g RSS, dried at a temperature of 135 °C for 3 h, was placed in a 100 mL capacity volumetric flask with and dissolved by heating in a water bath in 85 mL of 96% ethanol, then it was cooled and the solution volume was adjusted with the same alcohol up to the mark and mix (solution A) with RSS. Then, 2 mL RSS solution A, 1 drop of 5% acetic acid, and 5 mL of 2% alcohol solution of aluminum chloride were placed in a 25 mL volumetric flask and diluted with 96% ethanol to the mark (solution B) RSS [42].

### 2.4. Microbial Strain and Culture Medias

The study of biofilms and phenotypic features was carried out using the reference strain of *C. albicans* ATCC 10231 https://vkpm.genetika.ru/katalog-mikroorganizmov/cat30012009367/ (accessed on 2 November 2023). We chose this strain because we have worked with it for many years and it is already well characterized [7,10,12]. The strain was stored at −80 °C in a medium with glycerol [12].

The microorganisms were cultured for 24 h at 37 °C using the following media: Heart brain broth (HiMedia, Mumbai, Maharashtra, India), Sabouraud broth, and agar (BioMerieux, Marcy-l'Étoile, France). All the operational conditions were identical to those used in our prior article [43].

### 2.5. Biofilm Formation Ability

Biofilm formation of *C. albicans* ATCC 10231 and the effect of ATE on the suppression of biofilm formation were assessed using the standard microplate culture method. Microplates with 96 wells had a comfortable volume without pouring out—200 μL. YLF culture was grown in advance for 24 h at a temperature of 37 °C on the Sabouraud broth medium, in a volume of 10 mL, then centrifuged at $2.4 \times 10^3$ rpm, and a fungal suspension was prepared. In our work, we used the 2.0 McFarland turbidity standard (HiMedia, Mumbai, Maharashtra, India) with a densitometer DEN-1B (Biosan, Riga, Latvia). In the wells of a sterile polystyrene plate 100 μL of the Heart brain broth (Hbb) medium was added. Next, 200 μL of previously prepared ATE was added in the second empty well of the first row. The first well was left as a control. By successive transfer of 100 μL of the solution from the second well to the third, from the third to the fourth..., etc., we reduced the concentration of ATE by half each time [25].

Further, *C. albicans* inoculum was added in each well and cultivated 48 h at 37 °C. The microorganism culture was already in Hbb medium. This means that 75% of the volume of the well was always occupied by nutrition medium. And only 25% of the well was extracted in different concentrations. Nutrients were not depleted. They were always in the same quantity. The plate was cultured with the lid tightly closed. The progress of the experiment is presented in Table 1.

After cultivation, the microtiter plate was washed three times with phosphate buffer (PB) and dried for 15 min. A 1% solution of crystal violet was added to each well. The plate was incubated for 15 min, washed again with PB solution, and dried for 15 min. After that, 150 μL of 96% ethanol was added to each well, then incubated for 15 min and the result was recorded on the microplate reader at a wavelength of 492 nm [25,43,44].

**Table 1.** Stages of the study ATE effect on the *C. albicans* biofilm formation.

| | 1 | 2 | 3 | 4 | 5 | 6 | 7 | 8 | 9 | 10 | 11 | 12 |
|---|---|---|---|---|---|---|---|---|---|---|---|---|
| Action 1 | Hbb 100 µL | | Hbb 100 µL | Hbb 100 µL | Hbb 100 µL | Hbb 100 µL | Hbb 100 µL | Hbb 100 µL | Hbb 100 µL | Hbb 100 µL | Hbb 100 µL | Hbb 100 µL |
| Action 2 | | +ATE 200 µL | | | | | | | | | | |
| Action 3 | Not titrated | Transfer 100 µL | Transfer 100 µL | Transfer 100 µL | Transfer 100 µL | Transfer 100 µL | Transfer 100 µL | Transfer 100 µL | Transfer 100 µL | Transfer 100 µL | Transfer 100 µL | Transfer 100 µL |
| Titer of ATE | Control – no ATE | 1/1 | 1/2 | 1/4 | 1/8 | 1/16 | 1/32 | 1/64 | 1/128 | 1/256 | 1/512 | 1/1024 |
| Action 4 | +100 µL of culture | +100 µL of culture | +100 µL of culture | +100 µL of culture | +100 µL of culture | +100 µL of culture | +100 µL of culture | +100 µL of culture | +100 µL of culture | +100 µL of culture | +100 µL of culture | +100 µL of culture |
| Action 5 | Wait for 48 h | | | | | | | | | | | |

The average decrease was measured and used to calculate the biofilm inhibition percentage by ATE, where OD AS was the optical density average of *C. albicans* samples in the experiment and OD AC was the optical density average of isolates in control without ATE:

$$Average\ decrease\ OD(\%) = \frac{OD\ AS\ \times\ 100}{OD\ AC} - 100 \tag{2}$$

### 2.6. Morphological Characteristics of Candida Biofilms

The preparations were taken in a loop from the control and experimental wells, the drop was placed on a slide, fixed, stained with 1% solution of crystal violet, and examined under oil immersion using an optical microscope BIOMED MS-1 Stereo (Biomed, Moscow, Russia). Preparations for scanning electron microscopy with a MAIA3 model 2016 were completed without sputtering; the microscope is an ultra-high resolution SEM (Tescan, Brno, Czech Republic).

### 2.7. Statistical Analysis

Experiments were performed in triplicate and results are expressed as means ± standard deviation. To analyze the correlation dependence, we used the Spearman method correlation. All the results obtained were processed in the XLStats 2016 program. All the graphs were plotted using Microsoft Excel 2016 (Microsoft Excel for Office 365 MSO, Microsoft COP., Redmond, WA, USA).

### 3. Results

The content of flavonoids in terms of rutin was $1.896 \pm 0.071\%$ per 1 mL ATE. We carried out a double titration of the extract and expressed its concentration in Table 2 by the sum of flavonoids.

**Table 2.** Results of densitometric studies of the effect of different ATE concentrations on *Candida albicans* biofilm formation.

| | 1 | 2 | 3 | 4 | 5 | 6 | 7 | 8 | 9 | 10 | 11 | 12 |
|---|---|---|---|---|---|---|---|---|---|---|---|---|
| ATE concentration according to rutine | Control no ATE | 1.896 | 0.948 | 0.474 | 0.237 | 0.119 | 0.06 | 0.03 | 0.015 | 0.008 | 0.004 | 0.002 |
| Average OD of 3 repeating the experiment | 0.525 ± 0.022 | 0.123 ± 0.018 ** | 0.142 ± 0.024 ** | 0.160 ± 0.021 ** | 0.154 ± 0.017 ** | 0.185 ± 0.019 ** | 0.194 ± 0.015 ** | 0.202 ± 0.023 ** | 0.211 ± 0.018 ** | 0.369 ± 0.031 * | 0.383 ± 0.024 * | 0.498 ± 0.018 |
| Average decrease OD, % | 0 | 76.6 | 73.0 | 69.5 | 70.7 | 64.8 | 63.0 | 61.5 | 59.8 | 29.7 | 27.0 | 5.1 |

Note: * ($p \leq 0.05$), ** ($p \leq 0.01$)—level of statistical significance of biofilm optical density data compared to control.

As a result, *C. albicans* biofilm with an average OD value of 0.525 was formed in the first control well. This indicator characterizes our chosen microorganism as a very strong biofilm producer. A similar result has been reported in our published papers [3,7,10].

In wells 2-3-4-5-6-7-8-9, *C. albicans* biofilms either did not form at all, or (which is most likely) were washed away from the well after washing three times, since adhesion did not occur under the influence of ATE and fixing to the surface of the plastic. Starting from the 10th well, where the ATE concentration was low, the adhesion of microorganisms and the formation of a biofilm occurred. In well 12, the optical density of biofilms almost approached the control values.

As a result, we had questions. Is it only the concentration of ATE that matters in this experiment? Or is it also worth taking into account the exposure time to ATE? Or, perhaps, ATE does not allow biofilms to "emerge and develop". . . and it will have no effect on a ready-made, formed biofilm. . .

To answer these questions, we decided to conduct a second experiment in the following way. First, we developed a biofilm and then add ATE for several hours. We added 100 µL of *C. albicans* culture in Hbb at a concentration of 2.0 (McFarland) into each well of the first row with an automatic pipette. The total volume of the wells was 100 µL. The plate was cultured

at 37 °C for the same 48 h. Since YLF culture is in a comfortable environment—Hbb—and at a comfortable temperature—+37 °C—close to the temperature of the human body, it is absolutely normal that the microorganism forms biofilms within 48 h. But, since the volumes are very small, we tightly closed the plate with a lid and carefully wrapped the edges with electrical tape in order to prevent the medium from drying out.

After 2 days, it was necessary to add ATE at different concentrations. For this purpose, ATE titration was carried out on a separate plate. To avoid confusion, 1 well always remained the control, and it contained only 100 μL of medium. And, starting from the second well it was an experimental set. Then, the contents of the second plate were added to the first and cultured with the lid closed at 37 °C for another 6 h, i.e., the total incubation time became 48 + 6 = 54 h. The total volume of the wells was the same as in the first experiment: 200 μL. But in this case, we would be able to evaluate the effect of ATE on the already formed biofilm. Experiment 2 is presented in Table 3. Then, the procedures: washing, fixing, staining, and drying, and dye elution, were repeated as in the first experiment.

**Table 3.** Stages of the second study ATE effect on the formed *C. albicans* biofilms.

| | 1 | 2 | 3 | 4 | 5 | 6 | 7 | 8 | 9 | 10 | 11 | 12 |
|---|---|---|---|---|---|---|---|---|---|---|---|---|
| Action 1 | *C. albicans* in Hbb 100 μL | *C. albicans* in Hbb 100 μL | *C. albicans* in Hbb 100 μL | *C. albicans* in Hbb 100 μL | *C. albicans* in Hbb 100 μL | *C. albicans* in Hbb 100 μL | *C. albicans* in Hbb 100 μL | *C. albicans* in Hbb 100 μL | *C. albicans* in Hbb 100 μL | *C. albicans* in Hbb 100 μL | *C. albicans* in Hbb 100 μL | *C. albicans* in Hbb 100 μL |
| Action 2 | Wait 48 h | | | | | | | | | | | |
| Action 3 | +100 μL Hbb | + Not titrated ATE | + titrated ATE | + titrated ATE | + titrated ATE | + titrated ATE | + titrated ATE | + titrated ATE | + titrated ATE | + titrated ATE | + titrated ATE | + titrated ATE |
| Titer of ATE | Control – no ATE | 1/1 | 1/2 | 1/4 | 1/8 | 1/16 | 1/32 | 1/64 | 1/128 | 1/256 | 1/512 | 1/1024 |
| ATE concentration according to rutine | Control no ATE | 1.896 | 0.948 | 0.474 | 0.237 | 0.119 | 0.06 | 0.03 | 0.015 | 0.008 | 0.004 | 0.002 |
| Action 4 | Wait for 6 h | | | | | | | | | | | |

As a result of the second experiment, *C. albicans* biofilm with an average OD value of 0.653 was formed in the first control well. This can easily be explained by the fact that in the control the biofilm developed for 6 h longer than in the first experiment. After 48 h of incubation, we added fresh Hbb medium to the control and over the additional time the biofilm was able to "get stronger". In wells 2-3-4-5-6-7-8-9-10-11-12, YLF biofilms were formed and were not completely washed out of the well after washing three times. From this we can conclude that the action of ATE is largely aimed at the primary adhesion and fixation of young forming biofilms on the surface of the substrate (as in experiment 1). In the 12th well, the optical density of biofilms reached control values, which means that at such a low concentration (titer 1:1024) there is no effect of ATE (Table 4).

**Table 4.** Results of the second study: ATE effect on the formed *C. albicans* biofilms.

| | 1 | 2 | 3 | 4 | 5 | 6 | 7 | 8 | 9 | 10 | 11 | 12 |
|---|---|---|---|---|---|---|---|---|---|---|---|---|
| ATE concentration according to rutine | Control no ATE | 1.896 | 0.948 | 0.474 | 0.237 | 0.119 | 0.06 | 0.03 | 0.015 | 0.008 | 0.004 | 0.002 |
| Average OD of 3 repeating the experiment | 0.653 ± 0.025 | 0.323 ± 0.028 ** | 0.449 ± 0.020 * | 0.461 ± 0.019 * | 0.454 ± 0.021 * | 0.485 ± 0.019 * | 0.504 ± 0.025 * | 0.502 ± 0.026 * | 0.522 ± 0.018 * | 0.559 ± 0.023 * | 0.584 ± 0.025 * | 0.668 ± 0.028 |
| Average decrease OD, % | 0 | 50.5 | 31.2 | 29.4 | 30.5 | 25.7 | 22.8 | 23.1 | 20.1 | 14.4 | 10.6 | −2.3 |

Note: * ($p \leq 0.05$), ** ($p \leq 0.01$)—level of statistical significance of biofilm optical density data compared to control.

The morphological features of biofilms under the influence of ATE are presented in Figure 2.

In the control, the strain of *C. albicans* formed biofilms typical for this species. After 6 h of incubation, well-stained large (2.0–6.0 μm in diameter) cells are visible under a

microscope in the control. This is the so-called yeast phase; it is represented by budding cells of a round or oval shape. They are united with each other; darkening zones are occasionally visible between them—ECM. This is a typical initial stage of microorganism adhesion (Figure 2A). In the same well, but after 48 h of incubation, coaggregations of yeast and micellar form, united by ECM, and the presence of long branched hyphal forms, forming dense structures of pseudomycelium, is visible. This is also a characteristic picture for the architecture of a mature *C. albicans* biofilm, in accordance with the time and cultivation conditions. It is worth paying attention to how well the dye is perceived and how the coloring of the preparation occurs. It is not so much the ECM that is stained but the yeast and mycelial forms (Figure 2B).

The picture changes dramatically upon exposure to ATE. With the same method and time of staining, *C. albicans* cells appear to take up the dye differently: more of the ECM is stained than the cells themselves, which look like pale "shadows". YLF are represented by individual yeast cells; micellar forms are completely absent (Figure 2C,D). Why did *Candida* suddenly start to become more poorly stained in the presence of ATE? This means there were changes in the cell wall structure. Which structures were being influenced? The matrix, on the contrary, stains better. It seems like a certain component "comes out" from the cell walls into the surrounding space—into the matrix—which becomes more visible under a microscope. And the fact that the mycelial forms are not visible indicates that the constant presence of ATE in the environment blocks the development of hyphal forms.

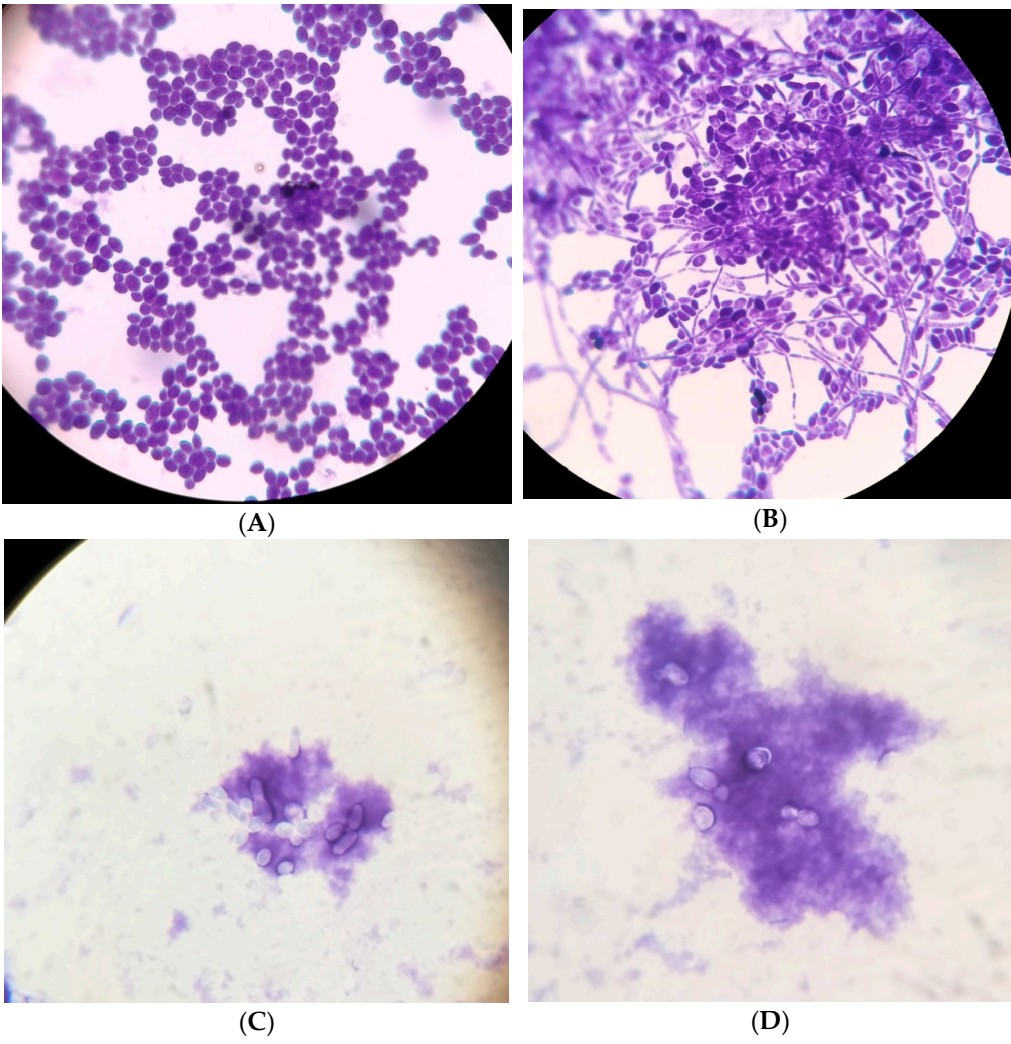

(A)

(B)

(C)

(D)

**Figure 2.** *Cont.*

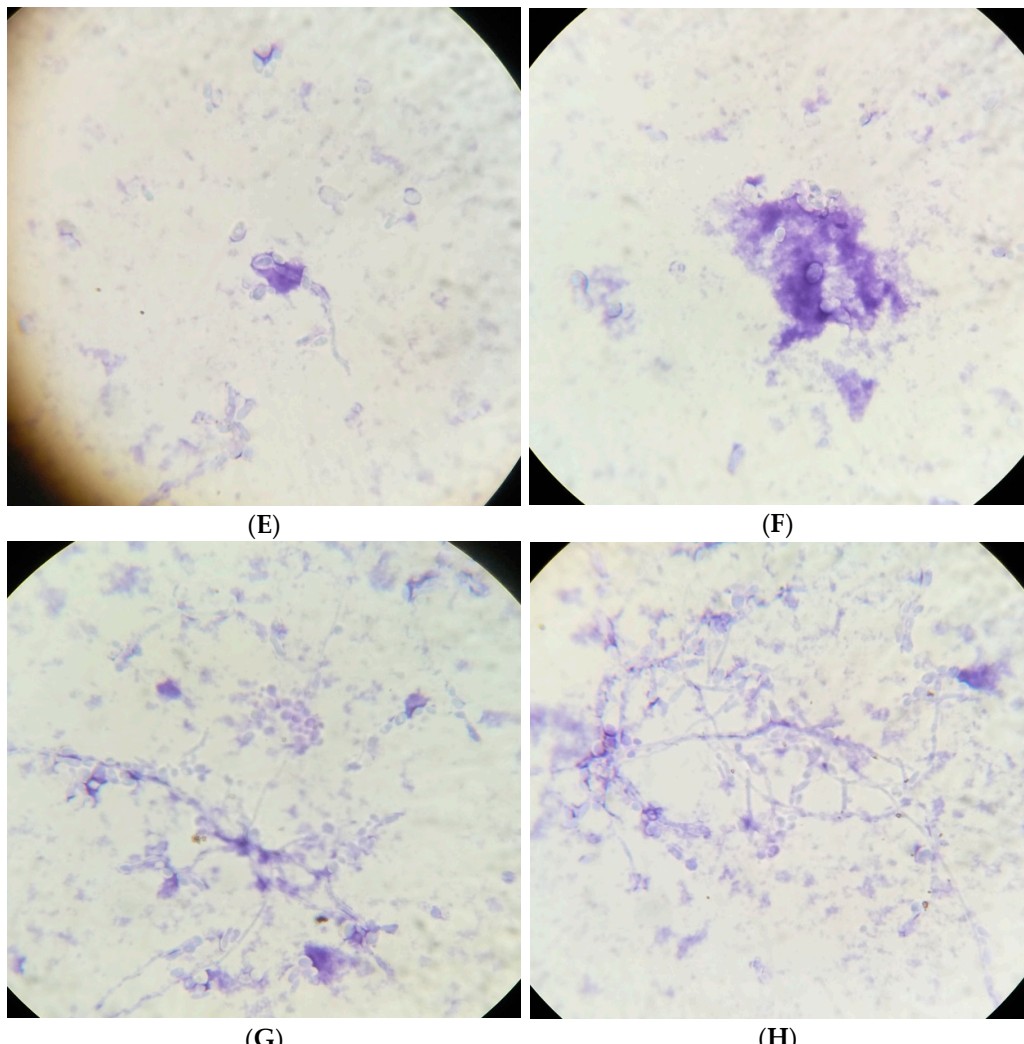

**Figure 2.** Morphology of yeast-like fungi. Staining with crystal violet solution, Magnification: 10 × 100, immersion. (**A**)—Study 1 Well 1 control after 6 h of incubation. (**B**)—Study 1 Well 1 control after 48 h of incubation. (**C**)—Study 1 Well 11 experiment after 48 h of incubation. (**D**)—Study 1 Well 12 experiment after 48 h of incubation. (**E**)—Study 2 Well 4 experiment after 48 h of incubation. (**F**)—Study 2 Well 5 experiment after 48 h of incubation. (**G**)—Study 2 Well 11 experiment after 54 h of incubation. (**H**)—Study 2 Well 12 experiment after 54 h of incubation.

In the second study, we first developed the biofilm and then exposed it to ATE. It is therefore not surprising to see the presence of long branched filaments of pseudomycelium in Figure 2G,H. However, again, the degree and nature of staining attracts attention. Unfortunately, electron microscopy cannot detect changes in ECM or fungal cells of the same size and shape in the experiment and control. In the control, they are located closer to each other, i.e., the degree of adhesion between cells is higher (Figure 3)

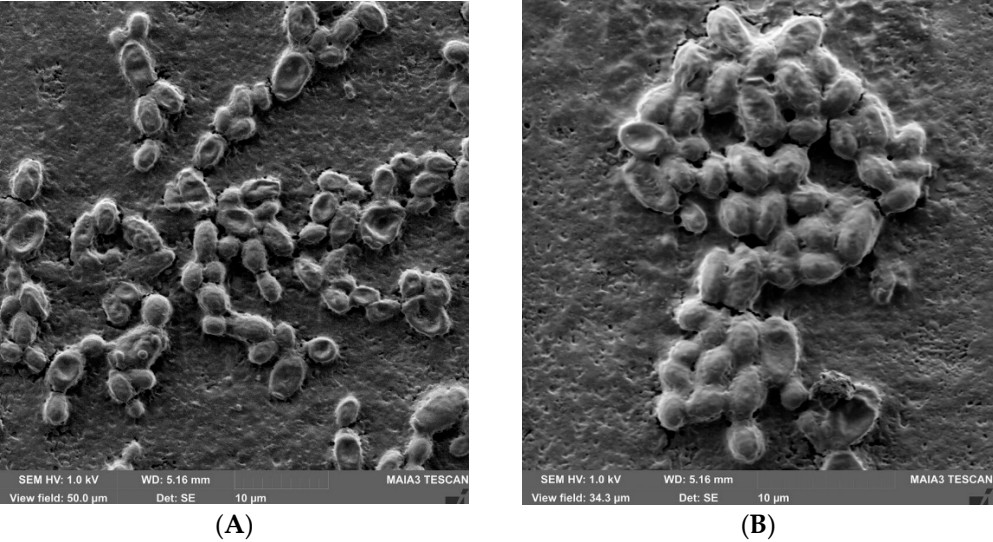

**Figure 3.** *C. albicans,* SEM. (**A**)—Study 1 well 2 experiment after 48 h of incubation. (**B**)—Study 1 well 1 control after 48 h of incubation. Magnification: 2000× MAIA3 Tescan (Czech Republic).

## 4. Discussion

The course of infectious diseases can occur with complications due to the formation of microbial biofilms in the body. Biofilms are a form of microbial community fixed on surfaces and consisting of microbial cells and an associated ECM. It is with biofilm infections that many chronic fungal and bacterial diseases are associated since they are better fixed on the cells of the human and animal body and are less amenable to antibiotic therapy. An incredible amount of research around the world is now aimed at finding antimicrobial agents with anti-biofilm activity. And plant extracts have always occupied a special position since their effectiveness has been proven in traditional medicine. It can also be noted that they are safer compared to synthetic drugs. Therefore, searches for phytoextracts will always be relevant in various aspects of medicine and pharmacy [22,45,46].

The design of this study was based on two types of experiments. The first one was based on the registration of morphological and functional changes in newly formed *Candida albicans* biofilms exposed to ATE. The second type of experiment was based on the registration of morpho-functional parameters (and their changes) of the action of ATE on ready-formed YLF biofilms.

The first experiment proved that ATE in concentrations from 1.896 to 0.015% according to rutin completely blocks the primary adhesion and fixation of *Candida albicans* on plastic surfaces, which is one of the main virulence factors of this pathogen. When the attaching ability of a microorganism is weakly expressed, its rapid removal from the body occurs. In our experience, such biofilms were completely washed off with three washings. The constant presence of ATE in the medium at a concentration of 0.008–0.004% significantly reduced the density of YLF biofilms and blocked the development of hyphal forms; led to loosening of the ECM; and changed the cell wall of fungi, which as a result absorbed the dye worse. Micro doses of the drug between 0.002 and below did not significantly affect *Candida* biofilms.

The second experiment proved that 1.896% ATE significantly reduced the density of *Candida albicans* biofilms; led to loosening of the ECM; changed the cell wall of *Candida*, especially strongly in hyphal forms, and as a result the dye uptake was poor. The formed biofilm under the action of the drug at 0.948–0.004 was partially destroyed and washed out. Micro doses of the drug from 0.002 and below did not affect the formed *Candida* biofilms at all.

## 5. Conclusions

We have long since moved beyond the first antimicrobial era, which involved optimizing the body's strengths to fight infection and the effects of the first antibiotics. We are now emerging from the second era, when pathogens "learned" to fight the drugs that had been killing them for more than half a century. The third era is the time of combinations, as we are looking for the possibility of combining drugs and taking into account the characteristics of the patient and the type and strain of the pathogen. Phytoextracts are one of the most promising areas of therapy. In our study, *Angelica ternata* extract can reliably suppress the formation of biofilms of pathogenic microorganisms *Candida albicans* up to 76.6%. The herbal medicine has such a significant effect on the initiation, adhesion, and development of biofilms. If the extract is added to a developed biofilm, it has a significant effect on the matrix. As a result, the "glue" between the cells is washed out and they are more easily washed away from the substratum to which they are attached. In this case, the optical density of the biofilm is halved (50.5%).

From this, it can be concluded that the studied ATE, of course, has antimycotic activity and that the action of ATE is more directed at the primary adhesion and fixation on the surface of the substrate of young, emerging *Candida* biofilms. We also proved the fact that the action of the *Angelica ternata* extract leads to the leaching and permanent curing of pre-formed biofilm, i.e., it can be assumed that ATE can exhibit both preventive and therapeutic effects.

**Author Contributions:** Conceptualization, Formal analysis, Investigation, N.S., A.K. and V.S.; Writing—Original draft, A.I. (Alfia Ibragimova) and E.N.; Writing—Review and editing, A.A. and R.M.; Data curation, Project administration, and Resources, A.I. (Aleksej Ignatiev), D.E. and V.E. Software and supervision, A.R. and R.S. All authors have read and agreed to the published version of the manuscript.

**Funding:** This paper was supported by the RUDN University Strategic Academic Leadership Program and Research project number D.2-A/N-2023 of the Department of Veterinary Medicine, Agrarian Technological Institute RUDN University (Moscow, Russia), under the direction of Vatnikov Y.A.

**Institutional Review Board Statement:** Not applicable.

**Informed Consent Statement:** Not applicable.

**Data Availability Statement:** Upon request, the data will be made available from the corresponding author.

**Conflicts of Interest:** The authors declare no conflict of interest.

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
