# Peer review of "The Effects of Angelica ternata Extract from Kyrgyzstan on the Formation of Candida albicans ATCC 10231 Biofilms"

_applsci, doi:10.3390/app132112042_

Round 1
Reviewer 1 Report
Comments and Suggestions for Authors
Review Comments
1) Title: The effects of the Angelica ternata extract from Kyrgyzstan on the formation of Candida albicans ATСС 10231 biofilms. Please change as : The effects of Angelica ternata extract from Kyrgyzstan on the formation of Candida albicans ATСС 10231 biofilms
2) Abstract - Line 17-18: In this study the therapeutic potential of Angelica ternata extract considered for biofilm-associated fungi Candida albicans. Please change as: the therapeutic potential of Angelica ternata extract was tested against biofilm-associated fungi Candida albicans.
3) Line 23: base: replace with base or well of the microplate OR substratum
4) Introduction - Line 28-29: The Candidiasis is frequently involved in surface-associated biofilm formation disease [1,2]. Please change as: Candidiasis is frequently involved in surface-associated biofilm formation disease [1,2].
5) Line 29-31: These networks have multifaceted interactions with the host. Biofilms may be considered as aggregates of microbes enmeshed in an extracellular matrix (ECM) consisting of multifarious polymeric components, forming a complex three-dimensional architecture on biotic and abiotic surfaces [3,4].
6) Line 32: ECM acts as a glue, surrounding and clasping together сell in biofilms. /…cells in biofilm
7) Line 24: Several studies of our team have shown that some herbal extracts have proven their effectiveness in vitro and in vivo [25-29]. Some are nanoparticles of the plant extracts. They may be accordingly mentioned.
8) Line 24: Are references 28 and 29 from the author’s team?
9) Line 58: Epizootological correct as epizootiological
10) Line 70: and we provide links to the articles that are closest to our topic. / We have provided…
11) Line 71: Considering all of the above we decide to investigate the effect of /Considering all of the above we decided to investigate the effect of…
12) Add latest (2022 and 2023) references on the antimicrobial activity of plant and spice extracts against Candida albicans and its poly microbial biofilms.
13) Materials and Methods - Line 100: The powder of the roots, at the tip of a knife, is drunk with urinary retention, pneumonia. This sentence is confusing so please rephrase.
14) Line 104: In the preparation of the AT extract please also include the drying and powdering of the plant material.
15) Was the AT extract analysed by HPLC/ LC-MS/GC-MS to detect the actives?
16) Is the culture resistant to any conventional antimycotic drugs? Was AST done?
17) Line 160-161: The first hole was left as a control. / The first well was … line 225: hole. Only one well had the control? Table 1, indicates otherwise.
18) Line 181: … 1% solution of crystalline violet… / crystal violet
19) Line 183: Preparations for scanning electron microscopy were prepared…/ Please avoid redundancy as in Preparations… were prepared.
20) Explain why samples were not gold sputtered for SEM?
21) Results - Line 201: similar result has been repeated in our published papers… / repeated to be replaced with reported
22) Line 219: we tightly closed the die with a lid,… / what is this die, do you mean the plate
23) Line -223-224: 1 well always remained the control…/ only 1 well or is it the first column. Were replicates not maintained?
24) Line 224-225: And starting from the second hole it’s an experience. This sentence is confusing. Please change the hole to well and experience to experimental or treatment set
25) Were any experiments done to check for cell viability/cell death?
26) Line 255: … architectonics of a mature C. albicans biofilm/ replace architectonics with architecture
27) Line 260: … cells perceive the dye completely/ replace perceive with take up the dye
28) In SEM the magnification of the images should be given.
29) Discussion - Line 302 and 307: … as a result they perceived the dye worse. Rephrase as: …as a result the dye uptake is poor.
30) Line 323: washed away from the base to which / replace base with substratum
31) Line 327: … young, emerging biofilms Candida. / … young emerging Candida biofilms.
32) Line 326-328: From this it can be concluded that the studied ATE, of course, has antimycotic activity 325 and the action of ATE is more directed at the primary adhesion and fixation on the surface of the substrate of young, emerging biofilms Candida. I.e., it can be assumed that ATE can exhibit both preventive and therapeutic effects.
Should include the preformed biofilm information here and then conclude otherwise this paragraph has only the information about the emerging biofilm.
Comments on the Quality of English LanguagePlease do a grammar check and revise the manuscript accordingly. Some of the grammar issues have been indicated in the reviewer's comments.
Author Response
We would like to acknowledge the valuable comments of the reviewers, that have contributed to improve the clarity of the manuscript significantly. We enclose an itemized list of replies to the comments raised and the changes proposed in the manuscript, which are highlighted in green color in the file.

Reviewer 2 Report
Comments and Suggestions for Authors
The article entitled “The effects of the Angelica ternata extract from Kyrgyzstan on the formation of Candida albicans ATСС 10231 biofilms” describes the investigation of the therapeutic potential of Angelica ternata extract considered for biofilm-associated fungi Candida albicans. The manuscript would be of general interest to the researchers of this field, but the manuscript lacks some basic information that the author should consider and incorporate in the present form of the manuscript. Here are some concerns that need to be addressed.
Some comments and corrections for authors:
1. The manuscript has some punctuation and grammatical errors and needs to be corrected (i.e., there must be comma before and in all mns when mention about over two parameters). Please check all parts of the mns.
2. Please use “mL” instead of “ml”.
3. I believe the paragraph related with the “supplement” is unnecessary.
Comments on the Quality of English LanguageThe manuscript has some punctuation and grammatical errors and needs to be corrected (i.e., there must be comma before and in all mns when mention about over two parameters). Please check all parts of the mns.
Author Response

(The authors gave the same response as above.)

Reviewer 3 Report
Comments and Suggestions for Authors
As a general assessment, the argument of the study is well done, but there are some missing experiments that must be included (ex. cytotoxicity tests) and some error that must be. Also, the obtained results are not sufficient to argue the assumptions made by the authors.
Materials and method
Plant collection – the title must be Plant collection and identification
Paragraf 99-103 – must be moved to Discussion section;
Line 153 – 163 - the technique of performing binary dilutions in Hbb seems incorrect to me, the medium being practically too diluted in the first wells (dilutions 1/2, 1/4...). The non-development of the biofilm is induced by cell death due to lack of nutrients and not by the presence of ATE!
Line 164 – the inoculum volume is too large relative to the medium volume. Generally, the volumetric ratio must be 1/10 = inoculum/medium
Results
This section must be reorganized and the results the results must be reinterpreted according to what was obtained, without assumptions not experimentally proven.
In the Materials and method section, the author present “2.3. Quantitative determination of the amount of flavonoids in ATE”, but in the results are completely missing!
Paragraph 212-230 and Table 3 - Reorganize for a better understanding of the experiment flow and move to Methods section;
Conclusions
The conclusions regarding your own study are too brief and do not emphasize the originality of the study.
Author Response

(The authors gave the same response as above.)
